FluxPyt: a Python-based free and open-source software for 13C-metabolic flux analyses

Desai Trunil S. 1 2
http://orcid.org/0000-0001-6139-1904 Srivastava Shireesh 1 2 shireesh@icgeb.res.in
1 Systems Biology for Biofuels Group, International Centre for Genetic Engineering and Biotechnology , New Delhi, Delhi , India
2 DBT-ICGEB Center for Advanced Bioenergy Research, International Centre for Genetic Engineering and Biotechnology , New Delhi, Delhi , India
Wilke Claus
Electronic publication date: 2018 Apr 27
Publication date: 2018
Volume: 6
Electronic Location ID: e4716
Received 2018 Mar 2; Accepted 2018 Apr 13
Copyright: © 2018 Desai and Srivastava
Copyright year: 2018
Copyright holder: Desai and Srivastava
License: This is an open access article distributed under the terms of the Creative Commons Attribution License, which permits unrestricted use, distribution, reproduction and adaptation in any medium and for any purpose provided that it is properly attributed. For attribution, the original author(s), title, publication source (PeerJ) and either DOI or URL of the article must be cited.
License URL: https://creativecommons.org/licenses/by/4.0/

Keywords: Metabolic labeling studies, Flux analysis, Elementary metabolite unit, Flux maps, Mass isotopomer distribution, Central carbon metabolism

Funding: Department of Biotechnology (DBT), Ministry of Science and Technology, India BT/PB/Center/03/ICGEB/2011-Phase II Council for Scientific and Industrial Research (CSIR), India This work was supported by the Department of Biotechnology (DBT), Ministry of Science and Technology, India, through the grant no. BT/PB/Center/03/ICGEB/2011-Phase II. Trunil S. Desai’s PhD fellowship is funded by the Council for Scientific and Industrial Research (CSIR), India. The funders had no role in study design, data collection and analysis, decision to publish, or preparation of the manuscript.

==============================
13C-Metabolic flux analysis (MFA) is a powerful approach to estimate intracellular reaction rates which could be used in strain analysis and design. Processing and analysis of labeling data for calculation of fluxes and associated statistics is an essential part of MFA. However, various software currently available for data analysis employ proprietary platforms and thus limit accessibility. We developed FluxPyt, a Python-based truly open-source software package for conducting stationary 13C-MFA data analysis. The software is based on the efficient elementary metabolite unit framework. The standard deviations in the calculated fluxes are estimated using the Monte-Carlo analysis. FluxPyt also automatically creates flux maps based on a template for visualization of the MFA results. The flux distributions calculated by FluxPyt for two separate models: a small tricarboxylic acid cycle model and a larger Corynebacterium glutamicum model, were found to be in good agreement with those calculated by a previously published software. FluxPyt was tested in Microsoft™ Windows 7 and 10, as well as in Linux Mint 18.2. The availability of a free and open 13C-MFA software that works in various operating systems will enable more researchers to perform 13C-MFA and to further modify and develop the package.

Introduction

Metabolic flux analysis (MFA) is an important component of metabolic engineering (Stephanopoulos, 1999). MFA measures the distribution of flux in the different metabolic pathways of an organism as well as identifies the intracellular flux changes associated with altered cellular response such as overproduction of a desired metabolite for biotechnological applications or understanding the altered pathways in diseases. The flux values from MFA can be used to constrain the solution space of the linear optimization problem in flux balance analysis to give a physiologically meaningful flux distribution (Kim & Reed, 2012; Desai & Srivastava, 2015).

Metabolic flux analysis methodology has evolved over years. Earlier studies employed an approach termed stoichiometric MFA which balanced the fluxes around the intracellular metabolites in a stoichiometric network (vanGulik et al., 2001; Srivastava & Chan, 2008). While it provided important insights into altered metabolism under different conditions, there are several limitations of stoichiometric MFA such as the inability to determine fluxes in reversible or parallel reactions (Wiechert et al., 2001). 13C-MFA, which employs 13C-labeled substrates can resolve fluxes through such reactions. For a typical 13C-MFA, the cells are fed with a labeled substrate or mixture of labeled substrates, labeled at one more carbon atoms. The cells are required to be at metabolically steady state, i.e., when the intracellular concentrations of metabolites do not change over time (Zamboni et al., 2009). This ensures that the differences in the labeling patterns are not due to temporal variations in the fluxes, but due to the differential activity of the pathways. Once the labeled substrate attains steady state distribution throughout the metabolic network (isotopic steady state), the relative abundances of the mass isotopomers of intracellular metabolites and proteinogenic amino acids are measured using NMR (Szyperski et al., 1999) or mass spectrometry to derive their respective mass isotopomer distributions (MIDs). The gas chromatography/mass spectrometry technique for measurement of MIDs of tert-butyldimethylsilyl derivatives of amino acids is well established. The fragmentation pattern produced by the derivatives is also well characterized and these ions are used to calculate the MIDs which are eventually used to calculate the fluxes (Antoniewicz, Kelleher & Stephanopoulos, 2007b; Leighty & Antoniewicz, 2013; Crown, Long & Antoniewicz, 2015; Gonzalez, Long & Antoniewicz, 2017). Zamboni et al. provide a detailed protocol for 13C-MFA (Zamboni et al., 2009). A suitable metabolic model and the MID data are then fed in software tools which estimate the flux distributions in reactions included in the MFA model.

Along with the development of experimental techniques for stationary MFA, the methodology to calculate flux distribution from the labeling data has also evolved over the years. An important development in this direction was the introduction of the elementary metabolite unit (EMU) framework (Antoniewicz, Kelleher & Stephanopoulos, 2007a). The EMU decomposition algorithm detects minimum number of isotopomer balances needed to simulate the observed labeling pattern which greatly reduces the number of variables without any loss of information, thus making it much more efficient than the isotopomer mapping matrices approach (Schmidt et al., 1997). Based on the measured fractional enrichments of metabolites, the metabolic network is decomposed into elementary reactions of the EMUs. These elementary reactions form the basis of system of equations representing the relation between metabolic fluxes and the observed isotopic labeling. Recent MFA software such as WUFlux (He et al., 2016), Metran (Antoniewicz, Kelleher & Stephanopoulos, 2007a) and OpenFLUX (Quek et al., 2009) employ the EMUs framework. Most software for 13C-MFA such as OpenFLUX (Quek et al., 2009), WUFlux (He et al., 2016), INCA (Young, 2014) and Metran (Antoniewicz, Kelleher & Stephanopoulos, 2007a) work either on the proprietary MATLAB platform or are not free and open source. Many also use the MATLAB statistics and optimization toolboxes which are proprietary. Thus, a truly free and open-source software is not available yet. Understanding the requirement for open source tools, Birkel et al. have recently developed jQMM, an open source Python library to perform 13C-MFA which uses the general algebraic modeling system (GAMS) to solve the optimization problem (Birkel et al., 2017). However, GAMS is a proprietary system for mathematical optimization. We believe that a truly open-source software in which all the components and packages are free and open source would make MFA more accessible to researchers and also encourage further development of 13C-MFA and related tools.

Materials and Methods

FluxPyt is written in Python (version 3.6.1) because Python is an open-source platform and has free, well-developed data handling packages with ability to perform the various tasks in 13C-MFA. In FluxPyt, the numerical and matrix operations are conducted using the packages NumPy and SciPy (van der Walt, Colbert & Varoquaux, 2011), and SymPy (Meurer et al., 2017). The non-linear optimization (see Eq. 3 below) is conducted using the package lmfit (Newville et al., 2014). Python was installed as Anaconda3 (version 3.4.4.0) distribution. Anaconda3 also acts as a package manager to handle the dependency and compatibility issues between the different packages necessary for the functioning of FluxPyt. FluxPyt was developed and tested using the Spyder integrated development environment (IDE) with the IPython console. FluxPyt was tested in Windows 7, Windows 10 and Linux Mint (18.2) operating systems (OS). Detailed information of packages in the Anaconda distribution is provided in Data S1. While FluxPyt is designed to be used by people with limited programming skills, the open source nature allows further modification by advanced users. The general framework of FluxPyt is shown in Fig. 1. Table 1 describes different modules present in FluxPyt. The MFA model, isotope correction data and substrate input data are written in a csv file which can be easily edited in a spreadsheet. The reactions are divided into different types, namely, irreversible (“F”), reversible forward (“FR”), reversible reverse (“R”), metabolite balancing reactions (“B”) and isotopomer balancing reactions (“S”). The metabolites which do not take part in isotopomer balancing are marked as “X” in the atom transition equations. The basis flux, which typically is the substrate uptake flux in a network, is marked with a “*”. The upper bounds of other reactions are limited to 15 times this basis flux. The reverse free fluxes cannot be determined as they have no natural upper limit (Quek et al., 2009). The substrate information and the natural correction data are read from separate csv files which must be named substrate_input.csv and corr_file.csv, respectively. The correction file also specifies the number of mass isotopomer measurements for every amino acid provided for fitting. This is used to extract the MID vectors of length equal to the length of MID vectors provided in the measurement data. For detailed information on creating an MFA model for FluxPyt, a user guide is provided in the supplemental material (Data S2).

Figure 1 The general framework of FluxPyt.

The user of the software provides the model definition and the mass isotopomer distribution (MID) data. The metabolite model is then calculated. The flux parameter optimization is then performed using the Levenberg–Marquardt algorithm to calculate the MIDs and the best-fit fluxes. The differences in the calculated vs. the measured MIDs are plotted. The flux maps of the calculated flux distribution is automatically generated based on the provided svg file.

Table 1 Description of modules in FluxPyt.

Module	Description	
main.py	Main driver module. Asks user inputs	
build_model.py	Parses csv model file. Generates metabolite and isotopomer models, stoichiometric matrix. Reads measured MIDs	
create_atm_transition_equations.py	Creates list of elementary reactions from atom transition equations	
input_substrate_emu.py	Reads substrate input file and calculates substrate MIDs required in the emu network	
mid_vec_gen.py	Generates correction vector for a given natural isotope distribution vector and number of atomic elements	
priliminary_fba.py	Generates initial feasible flux distribution using glpk as linear programming solver and sets feasible reaction bounds	
glpk_solve.py	Solves linear programming problems using glpk	
make_emu_networks.py	Makes EMU networks from elementary reactions and measured MIDs	
free_flux_null.py	Calculates null matrix and free fluxes	
solve_mfa_lm.py	Generates and solves non-linear optimization problem using lmfit	
solve_mid_networks.py	Solves EMU network to find calculated MIDS with respect to parameters of the non-linear problem	
mid_corr.py	Corrects calculated MIDs	
draw_flux_map.py	Draws flux map from optimization result and flux map template	
monte_carlo.py	Performs Monte-Carlo analysis	
bootstrap.py	For bootstrapping results from Monte-Carlo analysis	

The model is automatically parsed into metabolite and isotopomer models. The metabolite model is a mass balance model that satisfies the metabolic steady state condition, (1) S⋅v=0

where, S is the stoichiometric matrix and v is the flux distribution vector. For a given stoichiometric matrix, the flux distribution vector can be written as a function of free fluxes (vfree) as follows: (2) v=N⋅vfree

where N is the null space of stoichiometric matrix.

The isotopomer balance model is generated based on the EMU decomposition algorithm (Antoniewicz, Kelleher & Stephanopoulos, 2007a). It assumes an isotopic steady state and is used to solve the non-linear optimization problem using the free fluxes as the optimization parameters to minimize the sum of squared residues (SSR) as given in Eq. (3).

(3) Minimize[(MIDcalc−MIDmeasured)2Error2]

Levenberg–Marquardt algorithm implemented in the lmfit package is used to optimize the free flux parameters in the metabolite model. The free flux assignments are done based on the reduced row echelon form (rref) of the stoichiometric matrix. Before calculating the rref, the reactions are arranged in following order: [vrev→virrevvfree→vbasisvrev←]

vrev→ and vrev← are the forward and reverse reactions of reversible reactions; virrev are irreversible reactions; vfree→ are reactions that the user chooses to act as free fluxes; vbasis are reactions with measured or predetermined values. The arrangement of reactions in a specified order ensures that the measured fluxes are calculated as free fluxes. In addition, the user may provide a preference for certain reactions to be calculated as free fluxes (vfree→). The default number of iterations is 10, but this value can be changed by the user. The iterations are run with different random initial parameters so as to increase the chances of finding global minimum.

Flux standard deviations are calculated using the Monte-Carlo method. Multiple datasets of the measured MID values are generated using the mean and standard deviations of the respective MIDs. The default value for the number of datasets is 500, but can be changed by the user. Each dataset of the measured MID values is then used to solve the non-linear optimization problem (He et al., 2016). This results in multiple optimal solutions (500 solutions by default) for each reaction. From these values the 68% and 95% confidence intervals (CIs) are determined from the upper and lower 16% and 2.5% values, respectively, using the bootstrap method (He et al., 2016) as follows. One thousand samples of 500 values each (if the default number of samples for Monte-Carlo analysis is used) are used to calculate the required percentiles (2.5, 16, 50, 84 and 97.5 percentiles). The calculated percentiles from each sample are stored as a separate set. The median of each percentile value is reported as the upper or lower boundary of the respective CI. For example, the median 2.5 and 97.5 percentile values represent the lower and upper boundaries of the 95% CI and the median 16 and 84 percentile values represent the lower and upper boundaries of the 68% CI, respectively. The median of the 50 percentile values represents the median of the estimated fluxes.

Goodness of fit

Assuming that the model is correct and the data are without gross errors, the SSR has a χ2 distribution with number of degrees of freedom equal to number of measurements to be fitted (n) minus the number of independent parameters (p) (Antoniewicz, Kelleher & Stephanopoulos, 2006; Leighty & Antoniewicz, 2013). The acceptable range of SSR is between χα/22(n−p) and χ1−α/22(n−p), where α is the chosen significance level, for example, α = 0.05 for 95% CI.

Output files and visualization of the results

FluxPyt generates several result files during the MFA, a list of which is provided in Table S1. The data results, containing the calculated MIDs, the optimal flux distribution values and CIs are stored as separate .csv and .pckl files. FluxPyt provides graphical representation of results. The differences between the measured and the calculated MIDs of the best fit solution are automatically plotted. Flux map is automatically generated using a template flux diagram created using Inkscape. The template for flux map is a svg file that can be created using Inkscape. Templates for tricarboxylic acid cycle (TCA) and Corynebacterium glutamicum networks are provided in the additional data (Data S1) or can be downloaded from the FluxPyt project site. The user can easily modify the template based on their model of interest using the same drawing tool to make flux maps for their organism of interest. The users need to write the ID of the reactions at the appropriate position in the template where they want the flux value of that particular reaction to be printed (Data S1). The bootstrap results are plotted as box plots. The user can choose the specific reactions to be plotted. Additionally, the bootstrap output is stored as a pandas dataframe (McKinney, 2010).

Results

We employed the models and the MID data given in OpenFLUX (Quek et al., 2009) for initial evaluation of FluxPyt. Both the TCA cycle and the C. glutamicum central metabolic network (Data S1) were compared.

Preliminary validation with the TCA cycle model

The TCA cycle metabolic network (see Data S1 for a detailed description of the model as well as a list of metabolites) consisted of 18 metabolite reactions and three isotopomer balancing reactions (type “S” reactions). The network has two substrate input reactions, pyruvate (V01) and glutamate (V02) and one reversible reaction (represented as two separate reactions, V07 and V08). Three of the reactions (V16, V17 and V18) have measured values relative to pyruvate flux fixed at 1. The stoichiometric matrix has seven free variables. Rearranging the reactions as mentioned in the implementation section above enables the measured flux values to be calculated as free fluxes. The three measured free flux parameters were allowed to vary within three standard deviations of their means while the remaining three free flux parameters were allowed to vary between the maximum bounds as explained in the implementation section. The only remaining parameter was the flux through pyruvate uptake reaction which was fixed to 1. A total of 18 data points were fitted to six flux parameters, therefore the maximum acceptable SSR was 21. The number of iterations was set to 10. Flux analysis results from both software converged to a similar solution. The MIDs calculated by FluxPyt were the same as those calculated by OpenFLUX (Table 2). Very minor differences in the values of two fluxes were observed (Table 3). These could be due to the different solvers and algorithms used to solve the least squares fitting problem and linear system of equations (He et al., 2016). The flux map, which was automatically generated (see Methods for details) based on the template, is shown in Fig. 2. The Monte-Carlo analysis suggested tight CIs for the TCA model (Fig. 3). The reversible reaction spanned the entire flux range, as expected. The optimal values for all fluxes lay within 2.5 and 97.5 percentile range, i.e., the 95% CI (Table 4).

Table 2 Mass isotopomer distributions (MIDs) for the TCA cycle model as calculated by FluxPyt and OpenFLUX.

		FluxPyt	OpenFLUX	
VALX:11111	M+0	0.0298	0.0298	
M+1	0.2438	0.2438	
M+2	0.0589	0.0589	
M+3	0.4399	0.4399	
M+4	0.0292	0.0292	
M+5	0.1984	0.1984	
LYSX:111111	M+0	0.0344	0.0344	
M+1	0.1854	0.1854	
M+2	0.1677	0.1677	
M+3	0.2499	0.2499	
M+4	0.2074	0.2074	
M+5	0.0764	0.0764	
M+6	0.0788	0.0788	
ASPX:1111	M+1	0.0661	0.0661	
M+2	0.3532	0.3532	
M+3	0.2497	0.2497	
M+4	0.1590	0.1590	
M+5	0.1720	0.1720	
Sum of squared residual (SSR)	0.0002	0.0002	

Table 3 Comparison of the optimal flux values for the TCA cycle model as calculated by FluxPyt and OpenFLUX.

Reaction ID	Reaction formula	FluxPyt	OpenFLUX	
V01	PYR_EX -> PYR	1.00	1.00	
V02	GLU_EX -> AKG	0.20	0.20	
V03	PYR -> ACCOA + CO2 + NADH	0.62	0.62	
V04	ACCOA + OAA -> ICI	0.62	0.62	
V05	ICI -> AKG + CO2 + NADH	0.62	0.62	
V06	AKG -> 0.5 SUC + 0.5 SUC + CO2 + NADH + ATP	0.59	0.59	
V07	SUC -> OAA + FADH2 + NADH	2.62	2.61	
V08	OAA + FADH2 + NADH -> 0.5 SUC + 0.5 SUC	2.02	2.02	
V09	OAA -> PYR + CO2	0.30	0.30	
V10	PYR + CO2 + ATP -> OAA	0.48	0.48	
V11	2 NADH + O2 -> 4 ATP	1.21	1.21	
V12	2 FADH2 + O2 -> 2 ATP	0.30	0.30	
V13	O2_EX -> O2	1.51	1.51	
V14	CO2 -> CO2_EX	1.65	1.65	
V15	ATP -> ATPM	5.56	5.54	
V16	PYR -> PYR_B	0.20	0.20	
V17	AKG -> AKG_B	0.23	0.23	
V18	OAA -> OAA_B	0.15	0.15	

Figure 2 Flux map of the TCA generated by FluxPyt.

The fluxes for the reactions of the TCA cycle model are shown as generated by the FluxPyt. Net flux values are shown for reversible reactions.

Figure 3 Flux confidence intervals (CIs) calculated by Monte-Carlo method for selected reactions.

The boxes span 16 and 84 percentiles and whiskers span 2.5 and 97.5 percentiles. See the TCA model in Data S1 for full form of the reactions. (A) The flux CIs for the reactions V02, V06 and V09, (B) the flux CIs for the reactions V08, V11, V13 and V14.

Table 4 The 95% and 68% confidence intervals for the TCA network calculated by the Monte-Carlo analysis in FluxPyt.

Reaction ID	−95% CI	−68% CI	Median	+68% CI	+95% CI	
V01	1.00	1.00	1.00	1.00	1.00	
V02	0.19	0.19	0.20	0.21	0.21	
V03	0.59	0.60	0.62	0.64	0.66	
V04	0.59	0.60	0.62	0.64	0.66	
V05	0.59	0.60	0.62	0.64	0.66	
V06	0.57	0.58	0.59	0.60	0.60	
V07	1.30	1.74	2.59	4.17	9.41	
V08	0.72	1.15	2.01	3.58	8.81	
V09	0.29	0.29	0.30	0.31	0.32	
V10	0.46	0.47	0.48	0.49	0.50	
V11	1.17	1.18	1.21	1.24	1.26	
V12	0.29	0.29	0.29	0.30	0.30	
V13	1.46	1.48	1.50	1.53	1.56	
V14	1.57	1.60	1.65	1.70	1.74	
V15	5.33	5.41	5.53	5.66	5.76	
V16	0.17	0.18	0.20	0.22	0.23	
V17	0.20	0.20	0.23	0.26	0.26	
V18	0.12	0.12	0.15	0.18	0.18	

MFA of C. glutamicum metabolic network

The C. glutamicum model (Quek et al., 2009) consisted of 67 reactions, 42 metabolites and 25 free fluxes (see Data S1 for a detailed description of the model as well as a list of metabolites). The flux measurements for 18 reactions were included in the model. The 24 variable parameters were fit to the 29 data points of amino acid labeling pattern. Here too, the calculated fluxes were similar to those calculated by OpenFLUX (Table S2).

The SSR values for the measured vs. calculated MIDs were similar for FluxPyt (690) and OpenFLUX (701) (Table 5). However, the SSR values for both the software were higher than the maximum acceptable SSR value (11.07), suggesting that either the model used was incorrect or there were measurement errors. Use of a recent MID data (Shupletsov et al., 2014), reduced the SSR to 10.33 (Table 6), which is within the acceptable range (1.15–11.07). This showed that the MID data previously used had measurement errors. The Fig. 4 shows the flux map for the C. glutamicum model for this data (see Table S3) for the list of flux values. The CIs calculated with Monte-Carlo method are shown in Table S4.

Table 5 The MIDs for the C. glutamicum metabolic network calculated by FluxPyt and OpenFLUX.

		Measured MIDs (Quek et al., 2009)	FluxPyt	OpenFLUX	
Ala 260	M+0	0.5085	0.5096	0.5095	
M+1	0.3529	0.3537	0.3538	
M+2	0.1058	0.1062	0.1063	
Val 288	M+0	0.3455	0.3472	0.3477	
M+1	0.3983	0.3979	0.3978	
M+2	0.1845	0.1840	0.1838	
Thr 404	M+0	0.3330	0.3342	0.3341	
M+1	0.3764	0.3750	0.3758	
M+2	0.1957	0.1958	0.1957	
Asp 418	M+0	0.3343	0.3336	0.3334	
M+1	0.3732	0.3743	0.3750	
M+2	0.1955	0.1962	0.1961	
Glu 432	M+0	0.2469	0.2499	0.2493	
M+1	0.3648	0.3652	0.3657	
M+2	0.2412	0.2392	0.2396	
Ser 390	M+0	0.4497	0.4483	0.4486	
M+1	0.3576	0.3582	0.3581	
M+2	0.1428	0.1437	0.1435	
Phe 336	M+0	0.2712	0.2736	0.2736	
M+1	0.3816	0.3812	0.3814	
M+2	0.2282	0.2282	0.2283	
Gly 246	M+0	0.7407	0.7416	0.7416	
M+1	0.1845	0.1852	0.1852	
Tyr 466	M+0	0.2344	0.2356	0.2356	
M+1	0.3530	0.3560	0.3561	
M+2	0.2423	0.2448	0.2448	
Tre 361	M+0	0.0613	0.0618	0.0618	
M+1	0.6040	0.6059	0.6060	
M+2	0.2070	0.2071	0.2070	
SSR		690	701	

Table 6 The MIDs for the C. glutamicum measured by Shupletsov et al. (2014) vs. those calculated by FluxPyt.

		Measured	Simulated	
Ala 260	m	0.504	0.506	
m+1	0.352	0.354	
m+2	0.107	0.109	
Val 288	m	0.346	0.346	
m+1	0.397	0.397	
m+2	0.185	0.185	
m+3	0.056	0.056	
Thr 404	m	0.332	0.333	
m+1	0.373	0.375	
m+2	0.195	0.196	
m+3	0.073	0.074	
Asp 418	m	0.331	0.333	
m+1	0.373	0.375	
m+2	0.195	0.197	
m+3	0.074	0.074	
Glu 432	m	0.248	0.248	
m+1	0.364	0.366	
m+2	0.240	0.241	
m+3	0.103	0.103	
Ser 390	m	0.443	0.448	
m+1	0.353	0.358	
m+2	0.142	0.145	
m+3	0.049	0.050	
Phe 336	m	0.273	0.273	
m+1	0.381	0.379	
m+2	0.229	0.228	
m+3	0.087	0.087	
Gly 246	m	0.731	0.737	
m+1	0.183	0.189	
m+2	0.072	0.074	
Tyr 466	m	0.235	0.235	
m+1	0.355	0.354	
m+2	0.245	0.245	
m+3	0.112	0.112	
Trem 361	m	0.062	0.062	
m+1	0.605	0.605	
m+2	0.208	0.208	
m+3	0.097	0.097	
SSR			10.33	

Figure 4 A map of the C. glutamicum fluxes generated by FluxPyt.

The fluxes of the metabolic network of the C. glutamicum calculated using the data from Shupletsov et al. (2014) are shown on the metabolic network. The figure is generated automatically by the software.

Our results show that FluxPyt can be used for performing MFA with real metabolic models. It is capable of flux estimation when multiple labeled substrates are used (e.g., the TCA model uses three labeled substrates, namely 1-13C pyruvate, U-13C pyruvate and 1-13C Glutamate).

Discussion

Metabolic flux analysis is a powerful method to measure intracellular fluxes. However, all of the currently used software have some component that is proprietary and thus makes it impossible to conduct 13C-MFA without obtaining/purchasing a license. Most licenses do not allow altering the code, restricting further development. In designing this open-source software, our effort was to keep the interface simple and easy to use for scientists somewhat familiar with other MFA software. Therefore, the files for labeling-data preparation have been kept similar to the popular software OpenFLUX. The software can be run through different interfaces. The Anaconda command prompt may be convenient for people familiar with Python, while the Spyder IDE would appear more familiar to scientists familiar with MATLAB interface. The IPython console in Spyder offers a more interactive environment than the command prompt. A print of FluxPyt for performing MFA of TCA network running in the Spyder console is provided in Data S3.

The fluxes calculated by MFA are dependent on the network chosen and the measurements used. In case of the C. glutamicum metabolic network, it was found that the MID data used by Quek et al. (2009) had measurement errors giving unacceptably high SSR for the MIDs calculated. Use of more recent MID data (Shupletsov et al., 2014) significantly reduced the SSR to acceptable values.

At this point the software is released for use in Microsoft Windows and Linux OS. Similarly, the CI calculations currently use the Monte-Carlo method. The open-source nature makes it possible for advanced users to adapt it to other OS as well as add other methods for calculating CIs.

Conclusion

FluxPyt is free and open-source software for conducting stationary 13C-MFAanalyses with performance comparable to other available software. It can be used for flux estimation with experimental data of practical scale. Automatic generation of flux maps allows better visualization of results. This tool will be useful to researchers interested in stationary 13C-MFA studies.

Supplemental Information

Supplemental Information 1 The FluxPyt wheel, MFA model folders, conda environment yml file.

Click here for additional data file.

Supplemental Information 2 The FluxPyt user guide.

Click here for additional data file.

Supplemental Information 3 A print of IPython console for performing MFA of TCA model with FluxPyt.

Click here for additional data file.

Supplemental Information 4 A list of output files generated by FluxPyt.

Click here for additional data file.

Supplemental Information 5 The optimal flux values for the C. glutamicum network calculated by FluxPyt and OpenFLUX.

Click here for additional data file.

Supplemental Information 6 The flux distribution for the C. glutamicum metabolic network calculated by FluxPyt using the MID data from Shupletsov et al. (2014).

Click here for additional data file.

Supplemental Information 7 The confidence intervals (CIs) for the fluxes of the C. glutamicum network as calculated by FluxPyt.

Click here for additional data file.

Additional Information and Declarations

Competing Interests

Author Contributions

Data Availability

The authors declare that they have no competing interests.

Trunil S. Desai conceived and designed the experiments, performed the experiments, analyzed the data, contributed reagents/materials/analysis tools, prepared figures and/or tables, authored or reviewed drafts of the paper, approved the final draft.

Shireesh Srivastava conceived and designed the experiments, analyzed the data, contributed reagents/materials/analysis tools, prepared figures and/or tables, authored or reviewed drafts of the paper, approved the final draft.

The following information was supplied regarding data availability:

Sourceforge: https://sourceforge.net/projects/fluxpyt/

Operating system: Windows 10, Windows 7 and Linux; Programming language: e.g. Python (∼3.6). Other requirements: e.g. NumPy, SciPy, SymPy, lmfit, csvkit, glpk, pandas. License: BSD 3-clause. Any restrictions to use by non-academics: None.

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
