# Peer review of "FluxPyt: a Python-based free and open-source software for 13C-metabolic flux analyses"

_PeerJ, doi:10.7717/peerj.4716_

## Round 0.1 · original submission · Minor Revisions

Both reviewers generally like your paper but have some concerns regarding the presentation of the work.

Reviewer 1 ·

Basic reporting

The technical details on "How-To" use the FluxPyt is clear in general (Suppl File 2), however there is still a bit more description required of the output files and figures the program generates.

Another technical detail is the need of the folder name (where the models are stored) is needed - the authors mentioned that there is no need to enter this folder name in the path to run FluxPyt. However, at least, I needed to add the model directory as well to the path to run FluxPyt.

Some text in the draft needs to be re-written, i.e., lines 33-36, 44-46, 79-81, 187-188 etc to make it a bit more clear.

Experimental design

There are lot of .pckl files generated along with some standard txt, png and svg file formats. An additional supplementary file describing at least the most useful files will be helpful, for example, flux map is definitely useful for the end user and are there any other file formats and descriptions that needs to be mentioned about? The authors should walk through the order of the output files generated as the program runs through different phases.

Either there should be a bit more introduction to EMU (elementary metabolite units) method or at least the authors should direct the reader(s) to Antoniewicz et. al's paper [PMC1994654] that describes EMU in detail. I see the reference citing the above article, but there should be a bit more stress, as Antoniewicz et. al's paper gives a very comprehensive description of the method. Also, did the authors try to compare FluxPyt to Metran (as Metran is from Antoniewicz/Stephanopoulos lab)?

Flux standard deviation calculations are a bit unclear (lines 124-128) and might need a bit more explanation of the need and use of this method.

Would this program be useful to run on non-stationary metabolic fluxes datasets too? How about the experiments with multiple tracers? Did the authors compare these datasets (i.e., multiple tracer experimental data) to OpenFlux as well? Since EMU framework seems to reduce the computational time, compared to traditional isotopomer methods, the authors should at least benchmark datasets with multiple tracers.

Validity of the findings

Most of the programs like OpenFlux and Metran that are based on EMU framework, even though are free for academia and not-for-profit institutions, should still be run on MATLAB which is not free. FluxPyt is built on python framework and makes this MFA software, a truly open source. The program is relatively easy to use, however the authors did not make any comments on use with other operating systems, like *nix or macOS. They should clearly mention that it is not compatible for other OS, or if the work is in progress to accommodate other OS.

Additional comments

The figures generated, ease of program will help other labs to use this program. However if there is a bit more description on how the output files can be used, it will be even more helpful.

Reviewer 2 ·

Basic reporting

The paper presents a python based software for 13C metabolic flux analysis. Through two case studies and comparison with other software, the authors demonstrate their software is able to generate valid results quantifying cell metabolism. The language is in general clear, and the figures and tables are supportive to the conclusions.

Experimental design

Since this is a software paper, there is no experiment for this study.

Validity of the findings

The authors have confirmed their software can reproduce MFA results that have been published previously.

Additional comments

I have some questions for this software, and hope the authors can address these questions in their manuscript if necessary.
1. As a matter of fact, many researchers use python 2.7 for programming, do you know if your software can work well in both 2.7 and 3.4 releases.
2. If someone wants to use FluxPyt to study a cyanobacterium, will the flux map automatically create a Calvin cycle? Or the flux map template is only for heterotrophs?
3. It will be better if you can give a full list of metabolites and their abbreviations somewhere in the manuscript.

---

## Round 0.2 · accepted · Accept

Thank you for carefully addressing the reviewer comments.

#